# A Group Contribution Method for Predicting the Alkyl Ester and Biodiesel Densities at Various Temperatures

Luis Felipe Ramírez-Verduzco [ID]

Dirección de Investigación en Transformación de Hidrocarburos, Instituto Mexicano del Petróleo, Eje Central Lázaro Cárdenas #152, Col. San Bartolo Atepehuacan, Mexico City 07730, Mexico; lframir@imp.mx;
Tel.: +52-55-9175-8390

**Abstract:** Biofuels are an attractive alternative from polluting activities to low carbon ones. In this understanding, biodiesel has the potential to replace fossil diesel. Density is a relevant parameter of biodiesel to work out its quality. Density models lead to reducing the costly and time-consuming experimental measurements. We compiled two databases to prove a group contribution approach. For this purpose, the first database contained 1231 densities of 58 alkyl esters, while the second covered 696 densities of 16 pure biodiesel samples and 8 biodiesel blends. The group contribution method based on the molar volume was used to estimate the alkyl ester densities, while the mixing rule proposed by Kay was used to predict the biodiesel densities. The method developed here is easy to apply and provides excellent results, because an average absolute deviation of 0.29% was reached on the biodiesel density prediction.

**Keywords:** density; alkyl ester; biodiesel; group contribution method; mixing rule

## 1. Introduction

Energy management from sustainable sources has been rising because of oil reserve depletion and price increases. Fossil fuels contribute to global warming and air pollution. In contrast, renewable options are better suited to mitigate the environmental damage caused by fuels. An interesting opportunity is the use of biodiesel [1] because it represents an ecological fuel made by treating vegetable oils or animal fats of diverse raw materials [2]. As an illustration, in the transesterification reaction, triacylglycerides react with an alcohol to make biodiesel and glycerine [3]. Biodiesel is composed of fatty acid alkyl esters [1,2]. Biodiesel is miscible with diesel and acronyms used to stand for the biodiesel volume in the mixture is a typical practice. For instance, B20 refers to a blend that has a twenty percent volume of biodiesel [4]. Biodiesel can be utilized alone or mixed with diesel in diesel-powered vehicles to reduce emissions and particle levels. The above is without significant engine modifications [1,4].

Biodiesel is recognized as a worthwhile substitute because of its advantages over petroleum-based fuels. For example, because of its oxygen content, biodiesel reduces the soot produced by the engine. Further, most biodiesel samples have higher cetane than fossil fuels. Biodiesel has a high flash point, causing a low fire risk. Biodiesel is friendly to the environment due to being produced from biodegradable raw materials with low sulfur content. In addition, it lengthens the motor life and is safer to handle and transport. However, biodiesel has inconveniences compared to fossil diesel. Among them are its high cost, lower energy content, higher viscosity, and poor cold flow properties [5,6].

Several properties set up the biodiesel quality. Thus, the standards ASTM D6751 and EN 14214 contain the biodiesel specifications. The foregoing ensures its correct production, storage, transport, and handling [7]. Data for biodiesel thermophysical properties are required to characterize the biofuel and optimize engine performance. Furthermore, the knowledge of biodiesel properties is important for the development, optimization, and control of processes and bio-refineries.

Density is one of the most significant properties of biodiesel for a correct formulation to satisfy quality requirements. So, its knowledge is valuable in the design and development of combustion models. Density, together with surface tension and viscosity, plays an important role in the atomization process for the engine injection system, affecting the size of fuel droplets and jet penetration. In addition, modest variations in density could influence the diesel power output because fuel injection is measured by volume, so given a certain volume injected, changes in density imply variations of mass injected into the combustion chamber [8,9].

Various researchers have developed different models to predict the density of both alkyl esters [10–12] and biodiesel [8,13,14] in the last decades. Although the literature presents various prediction model approaches to estimate the density of biodiesel and biodiesel blends, there will always be a need for new models to better characterize the biofuel density with greater accuracy compared to existing models.

Diverse efforts have been made to predict the properties of alkyl esters [15–20]. However, according to the best of our knowledge, only one work reports the prediction of the biodiesel density using a group contribution approach [8]. In that work, Pratas et al. measured the biodiesel density of 10 samples at temperatures from 278.15 to 373.15 K. Those experimental data were used by Pratas et al. along with other literature data to test the predictive capability of the model using Kay's mixing rule and the group contribution method based on the molar volume (GCVOL), getting an average deviation of 0.3% [8]. The work of Pratas et al. represents an important advance in the prediction of biodiesel density. Unfortunately, the number of data considered by Pratas et al. was low for biodiesel density. In addition, they did not highlight a comparison between experimental and calculated densities of alkyl esters. Thus, there is a necessity to extend and update the database to better prove the predictive capability of the model based on Kay's mixing rule and the GCVOL group contribution model.

In this work, we will compile a significant number of experimental densities for both alkyl esters and biodiesel samples. Furthermore, new GCVOL parameters that are ad hoc with the type of compounds studied will be proposed. Finally, we will set up a complete comparison of experimental and calculated densities, covering a wide temperature range, thus testing robustly the predictive capability of the model.

## 2. Methods

### 2.1. Experimental Database

We compiled two databases, one for the alkyl ester densities and one for the biodiesel densities. Each of them is described below.

A total of 1231 experimental densities reported in the literature [11,12,21–34] were compiled in this work, comprising 58 alkyl esters of between 7 to 25 carbon atoms. The densities were from 0.72358 to 0.9236 g/cm$^3$ at temperatures between 278.15 to 453.15 K (Table S1 of the Supplementary Materials). This compilation includes both saturated and unsaturated alkyl esters of different aliphatic chain lengths, with the unsaturated compounds having one, two, three, or four double bonds. Table S1 summarizes the number of experimental points compiled for each alkyl ester, including its temperature and density range encompassed during the experimental measurements.

Table S2 of the Supplementary Materials contains the fatty acid methyl ester composition of some biodiesel samples from the literature [8,35–50]. Overall, the compilation corresponds to 29 pure biodiesel samples and 34 biodiesel blends, representing 63 distributions of the methyl ester composition on biodiesel. In addition, Table S3 of the Supplementary Materials presents the information for the 696 experimental points on density collected in this work. Those data are in a range of 273.15 to 373.15 K and 0.8093 to 0.8961 g/cm$^3$ on temperature and density, respectively.

### 2.2. Predictive Model for the Alkyl Ester Density

We can estimate the liquid density of compounds using Equation (1) [8,51,52].

$$\rho = \frac{MW}{V} \tag{1}$$

where $\rho$ is the density expressed in g/cm$^3$. *MW* is the molecular weight expressed in g/mol. *V* is the molar volume expressed in cm$^3$/mol.

Regarding the group contribution method, it comprises dividing any compound into fragments, atoms, bonds, or groups of atoms, where each group has a partial value of a property called a contribution. By adding the contribution of each group, it is possible to calculate the compound physical property [53]. For this purpose, experimental data are usually used to estimate those contributions by fixing the parameters of each group.

Researchers have developed various models based on the group contribution principle in the last decades which differ in their applicability and the set of experimental data.

We can estimate the molar volume using Equations (2) and (3) and the GCVOL group contribution method [8,51,52].

$$V = \sum_i n_i \Delta v_i \tag{2}$$

where $n_i$ is the number of the *i-th* group, while $\Delta v_i$ is the *i-th* group contribution for the molar volume expressed in cm$^3$/mol. The value of $\Delta v_i$ can be estimated using Equation (3) [8,51,52].

$$\Delta v_i = A + BT + CT^2 \tag{3}$$

where *T* is the temperature expressed in K, while *A*, *B*, and *C* are the group contribution parameters which are expressed in cm$^3$/mol, cm$^3$/mol·K, and cm$^3$/mol·K$^2$, respectively.

The *A, B,* and *C* constants in Equation (3) can be adjusted by minimizing the objective function ($F_{obj}$), which is expressed in terms of the quadratic difference between the experimental and calculated densities [54]. A multiple regression based on the least-squares method can be used to adjust the parameters. The objective function, $F_{obj}$, is given by Equation (4).

$$F_{obj} = \sum_i \left( \rho_{exp,i} - \rho_{cal,i} \right)^2 \tag{4}$$

### 2.3. Predictive Model for the Biodiesel Density

Kay's mixing rule [55] can be used for estimating the density of a mixture and the above by using the properties of the pure components. Thus, Equation (5) presents the linear form of Kay's rule for density.

$$\rho_{biodiesel} = \sum_i c_i \rho_i + \sum_i \sum_j c_i c_j G_{ij} \tag{5}$$

where $\rho_{biodiesel}$ and $\rho_i$ correspond to the biodiesel and alkyl ester density, respectively. The composition of the alkyl ester is represented by $c_i$, expressed in mole fraction ($x_i$) or mass fraction ($w_i$), while $G_{ij}$ is the binary interaction parameter. $G_{ij}$ is usually zero or negligible when the compounds belong to the same chemical nature, as well as compounds with molecular weights close to each other. Clearly, this is often the case with biodiesel.

An alternative method for estimating the properties of the mixture is using Equation (6), where all the binary group contributions are substituted by a single correction factor.

$$\rho_{biodiesel} = \sum_i c_i \rho_i + F \tag{6}$$

where *F* is the correction factor expressed in g/cm$^3$.

### 2.4. Statistical Analysis

We used three statistical parameters to test the predictive capability of models and the above for both alkyl ester and biodiesel density. Those were the average absolute deviation (*AAD*), the correlation coefficient (*R*), and the standard deviation ($\sigma$), which can be estimated through Equations (7)–(9).

$$AAD = \frac{\sum_i \left( \left| \frac{(\rho_{exp,i} - \rho_{cal,i}) \times 100}{\rho_{exp,i}} \right| \right)}{n} \quad (7)$$

where $n$ is the number of experimental or calculated points, while $AAD$ is the average absolute deviation expressed in percentage.

$$R = \frac{\sum_i (\rho_{exp,i} - \overline{\rho_{exp}})(\rho_{cal,i} - \overline{\rho_{cal}})}{\sqrt{\sum_i (\rho_{exp,i} - \overline{\rho_{exp}})^2 \sum_i (\rho_{cal,i} - \overline{\rho_{cal}})^2}} \quad (8)$$

where $\rho$ with a top bar is the average density, and $R$ is the correlation coefficient. If $R$ is near to one is a sign that the model has good accuracy.

$$\sigma = \sqrt{\frac{\sum_i (\rho_{exp,i} - \rho_{cal,i})^2}{m - p}} \quad (9)$$

where $m$ is the number of experimental points, and $p$ is the number of parameters.

## 3. Results

We used 1173 experimental densities of the 1231 values reported in Table S1 to adjust the group contribution parameters, while we randomly selected and reserved 58 points to carry out the validation of the model. Appendix A contains some examples to illustrate the assignment that we made for the number of contribution groups present in the alkyl esters.

The GCVOL parameters bought in this work are presented in Table 1. We chose the Excel Solver tool for the optimization process. Furthermore, we employed the "Generalized Reduced Gradient" (GRG) method to determine the gradient or slope of the "Objective Function". Through this method, the input values changed until the partial derivatives were equal to zero, which implies that an optimal solution was reached. Certainly, the variation of parameters $A$, $B$, and $C$ in positive and negative values were allowed during the optimization process.

Table 1 contains the group contribution parameters reported by Elbro et al. [51], Ihmels and Gmehling [52], and Pratas et al. [8]. In addition, Table 1 contains the statistical parameters found with each approach for the 1173 data points.

**Table 1.** Group contribution and statistical parameters (density models for alkyl esters).

| Group | Group Contribution Parameters | | | Statistical Parameters | | |
|---|---|---|---|---|---|---|
| | $A$ (cm$^3$/mol) | $10^3 \cdot B$ (cm$^3$/mol·K) | $10^5 \cdot C$ (cm$^3$/mol·K$^2$) | $R$ | $AAD$ (%) | $\sigma$ (g/cm$^3$) |
| | This work | | | | | |
| -CH$_3$ | 15.74 | 1.62 | 10.01 | | | |
| -CH$_2$ | 14.42 | 5.1 | 0.76 | 0.9941 | 0.36 | 0.00339 |
| =CH- | 11.98 | 1.19 | 0.89 | | | |
| -COO- | 30.77 | 1.31 | 1.08 | | | |
| | Elbro et al. [51] | | | | | |
| -CH$_3$ | 18.96 | 45.58 | - | | | |
| -CH$_2$ | 12.52 | 12.94 | - | 0.9788 | 0.83 | 0.00866 |
| =CH- | 6.761 | 23.97 | - | | | |
| -COO- | 14.23 | 11.93 | - | | | |
| | Ihmels and Gmehling [52] | | | | | |
| -CH$_3$ | 16.43 | 55.62 | - | | | |
| -CH$_2$ | 12.04 | 14.1 | - | 0.9894 | 0.44 | 0.00457 |
| =CH- | −1.651 | 93.42 | −14.39 | | | |
| -COO- | 61.15 | −248.2 | 36.81 | | | |

**Table 1.** *Cont.*

| Group | Group Contribution Parameters | | | Statistical Parameters | | |
|---|---|---|---|---|---|---|
| | $A$ (cm$^3$/mol) | $10^3 \cdot B$ (cm$^3$/mol·K) | $10^5 \cdot C$ (cm$^3$/mol·K$^2$) | $R$ | $AAD$ (%) | $\sigma$ (g/cm$^3$) |
| | Pratas et al. [8] | | | | | |
| -CH$_3$ | 18.96 | 45.58 | - | | | |
| -CH$_2$ | 12.52 | 12.94 | - | 0.9911 | 0.81 | 0.00827 |
| =CH- | 11.43 | 6.756 | - | | | |
| -COO- | 14.23 | 11.93 | - | | | |

Figure 1 illustrates the relative error between the experimental and calculated densities, depending on the group contribution parameters used for predictions.

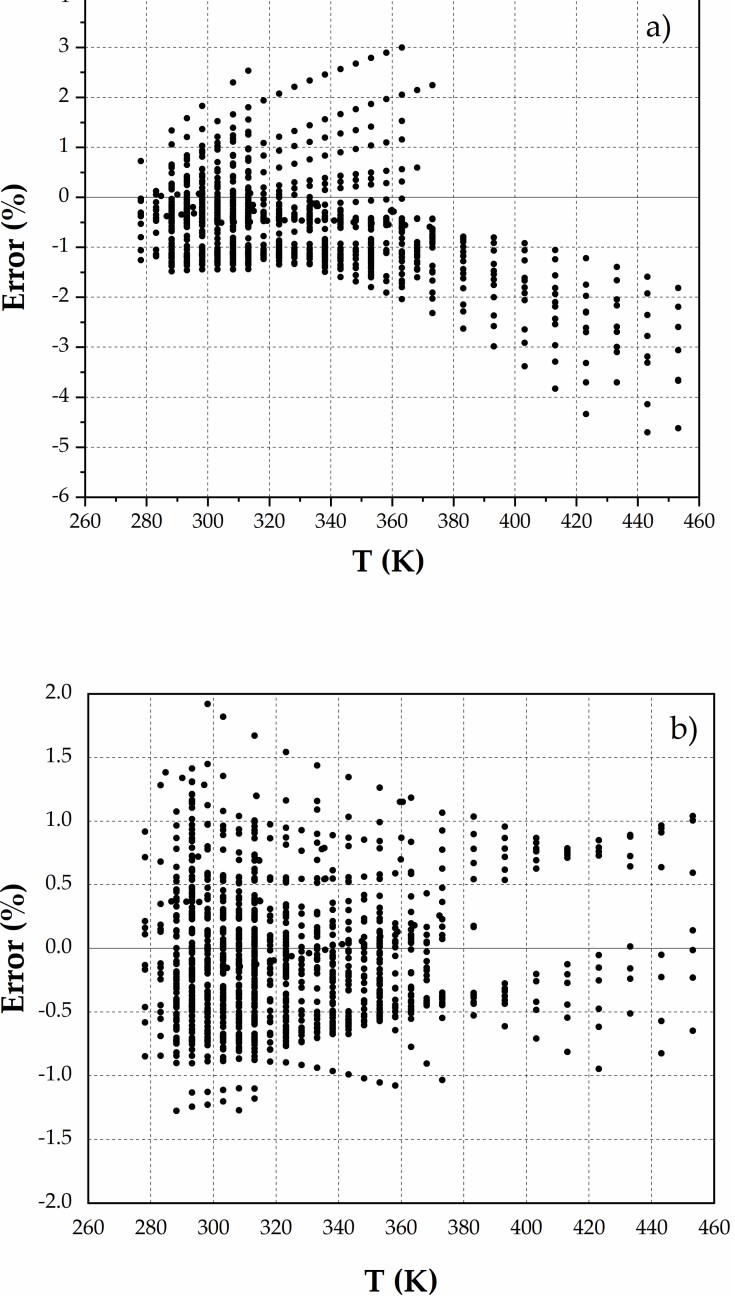

**Figure 1.** *Cont.*

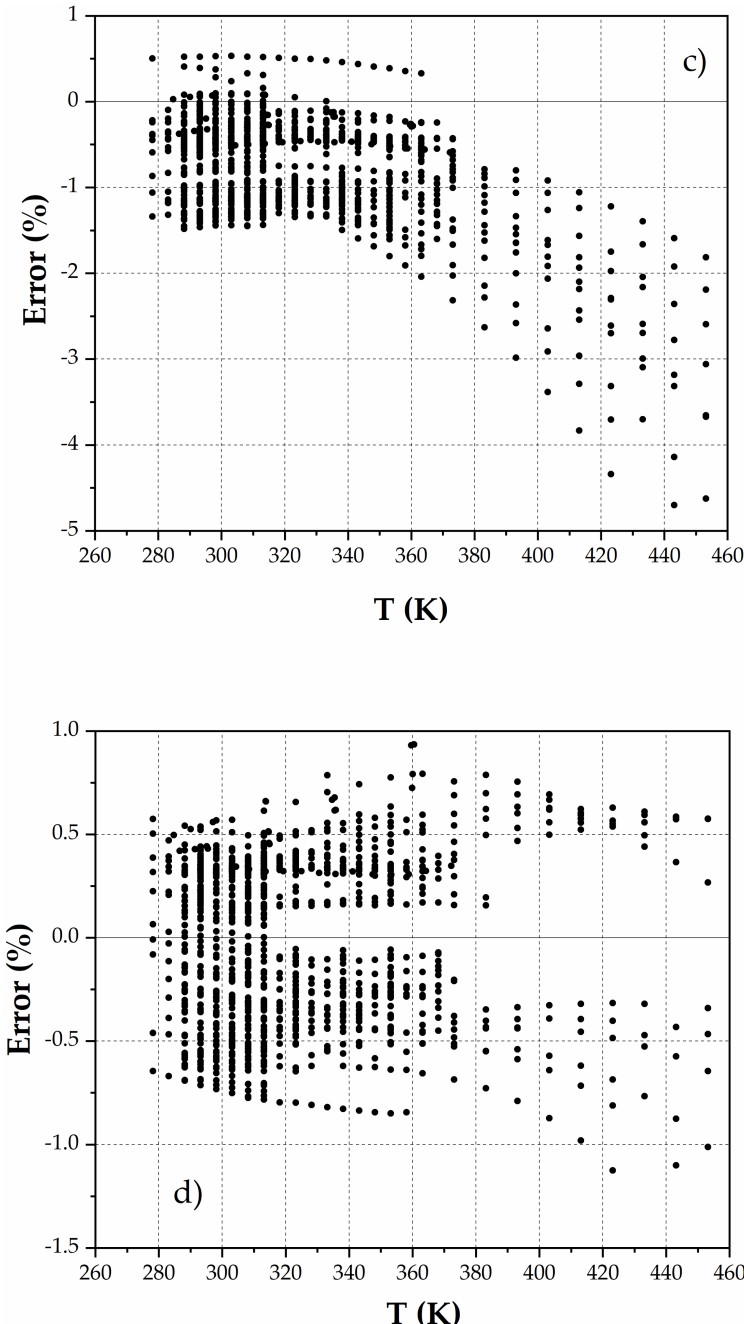

**Figure 1.** Relative error for the alkyl ester density. (**a**) Elbro, (**b**) Ihmels and Gmehling, (**c**) Pratas, (**d**) Our work.

Concerning the validation of our parameters, we obtained the following statistical parameters: $R = 0.9956$, $AAD = 0.4\%$, and $\sigma = 0.00392$ g/cm$^3$. Those statistical parameters were estimated based on the comparison between 58 experimental densities with the calculated ones.

Figure 2 exhibits some examples of the density variation with temperature for some alkyl esters, as well as the prediction capability of the method revised here. The experimental points are shown by symbols. The straight lines represent the estimates made with the group contribution method using the parameters adjusted in this work. Examples of some methyl esters are shown in Figure 2a, while those for some ethyl esters are illustrated in Figure 2b.

The model based on the group contribution was compared with two approaches available in the literature. The first comprises the use of the empirical correlation reported

in previous work [56] and the second method comprises predicting the density through an equation of state. For the latter, we used the well-known method described in the literature [57]. The values of critical temperature ($T_c$), critical pressure ($P_c$), and acentric factor ($w$) were extracted from Evangelista's work [19]. We test the alternative approaches using 1173 experimental data points. The results for the comparison between models are presented in Table 2.

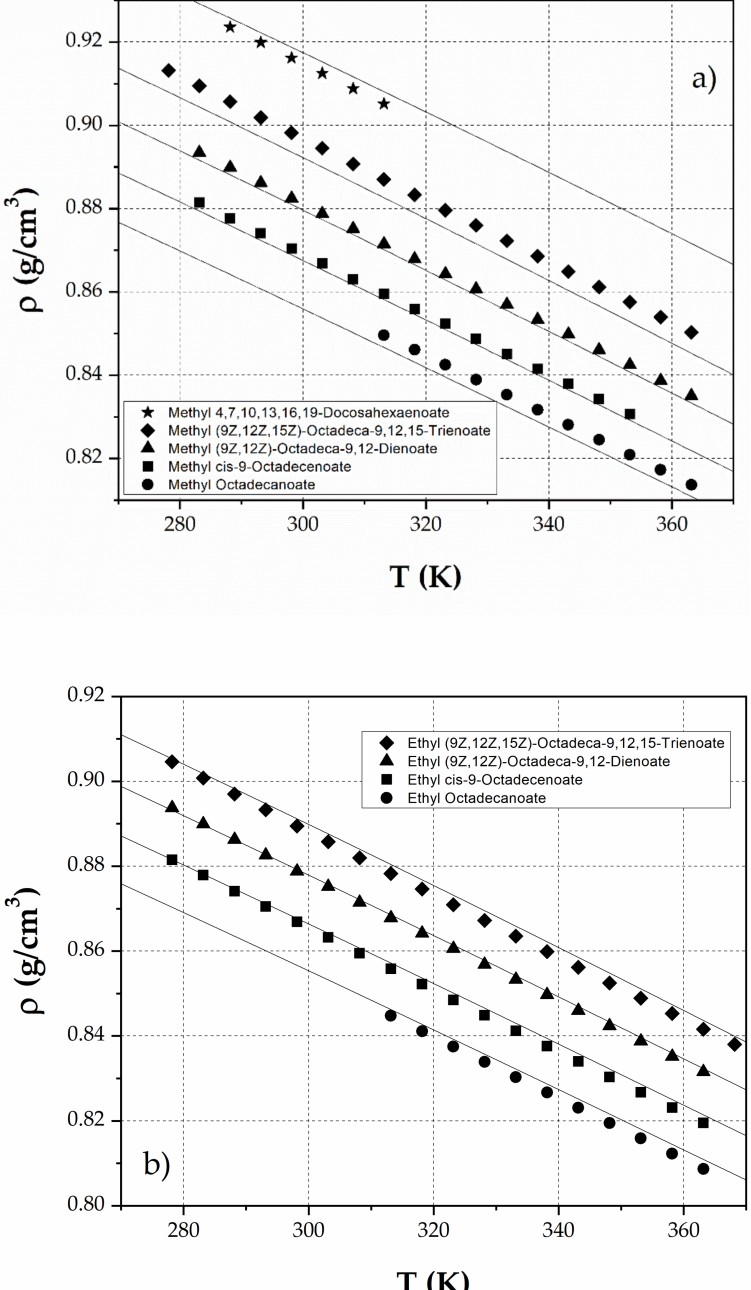

**Figure 2.** Density vs. temperature for some alkyl esters. (**a**) Methyl esters, (**b**) ethyl esters.

**Table 2.** Statistical parameters for the prediction of alkyl ester densities.

| Approach | R | AAD (%) | σ (g/cm³) |
|---|---|---|---|
| Empirical equation [56] | 0.987 | 0.55 | 0.00677 |
| Equation of State (EoS) [57] | 0.6139 | 7.59 | 0.0946 |
| GCVOL of this work | 0.9941 | 0.36 | 0.00339 |

The results for the biodiesel density predictions are shown in Table 3. The first three results were obtained by Equation (5) considering $G_{ij} = 0$ and using the group contribution parameters reported by each author [8,51,52], while the last result was obtained using Equation (6) considering the parameters estimated in this work and using $FC = 0.0056$ g/cm³.

**Table 3.** Statistical parameters for the prediction of biodiesel densities.

| Group Contribution Parameters Origin | R | AAD (%) | σ (g/cm³) |
|---|---|---|---|
| Elbro et al. [51] | 0.9807 | 0.65 | 0.00692 |
| Ihmels and Gmehling [52] | 0.9829 | 0.47 | 0.00525 |
| Pratas et al. [8] | 0.9838 | 0.29 | 0.00363 |
| This work | 0.9827 | 0.29 | 0.00371 |

Figure 3 depicts the relative error for the biodiesel densities for the approaches tested in this work.

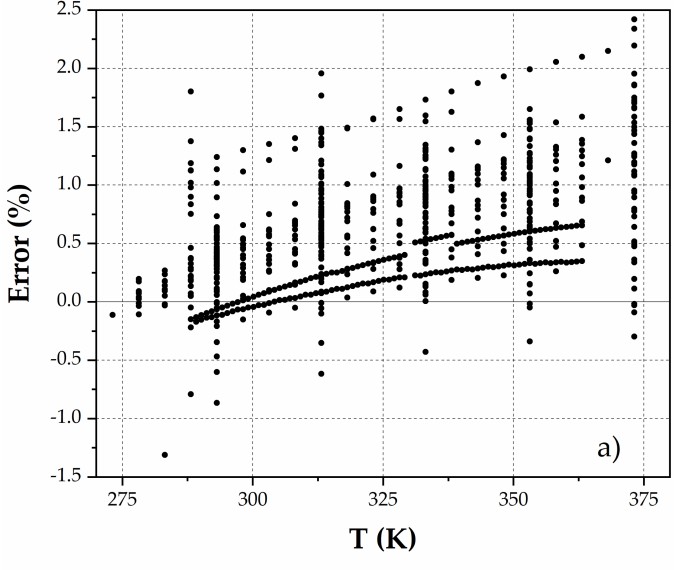

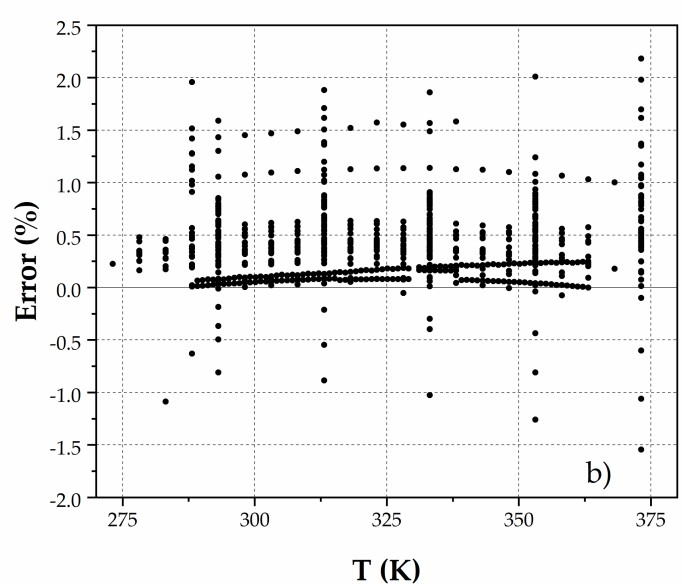

**Figure 3.** *Cont*.

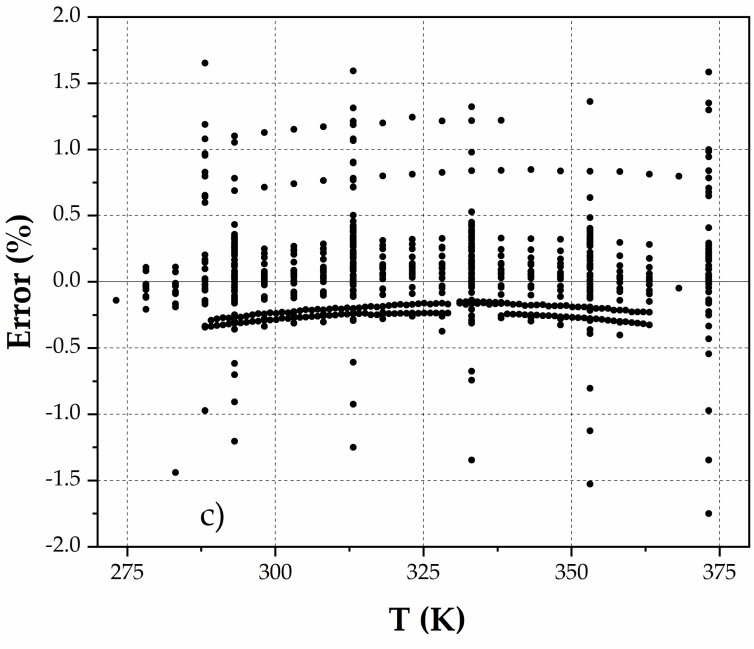

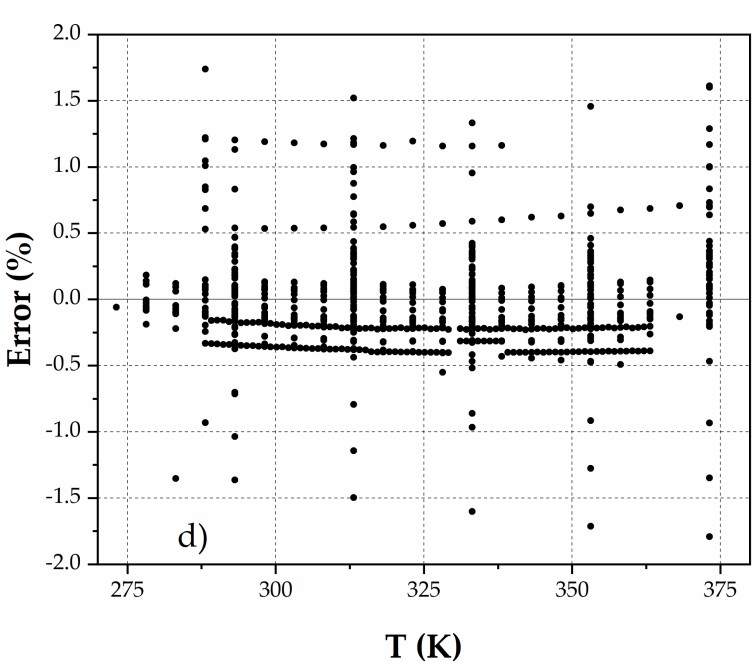

**Figure 3.** Relative error for the biodiesel density. (**a**) Elbro, (**b**) Ihmels and Gmehling, (**c**) Pratas, (**d**) Our work.

Figure 4 shows the narrow range of density variation for the different biodiesel samples studied in this work. This interval of values is contained by the area delimited by the straight lines. The experimental points examined in this work are represented by the black circles.

Figure 5 portrays a comparison of the prediction capability of the method to estimate the biodiesel density of some samples at various temperatures. The symbols represent the experimental information of the density of some biodiesel samples (produced from sunflower, cotton, palm, and coconut). Predictions made with Kay's mixing rule Equation (6) are represented by straight lines.

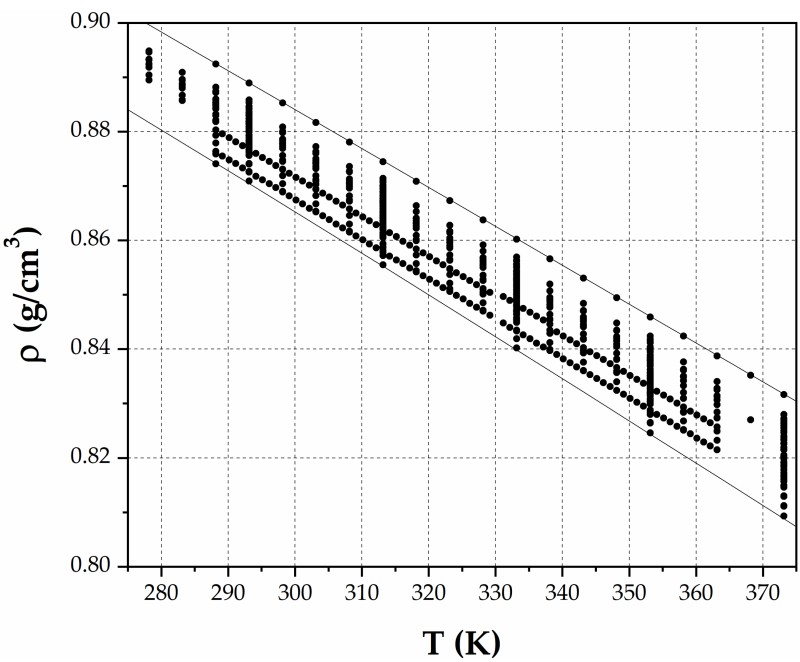

**Figure 4.** Range of density for various biodiesel samples.

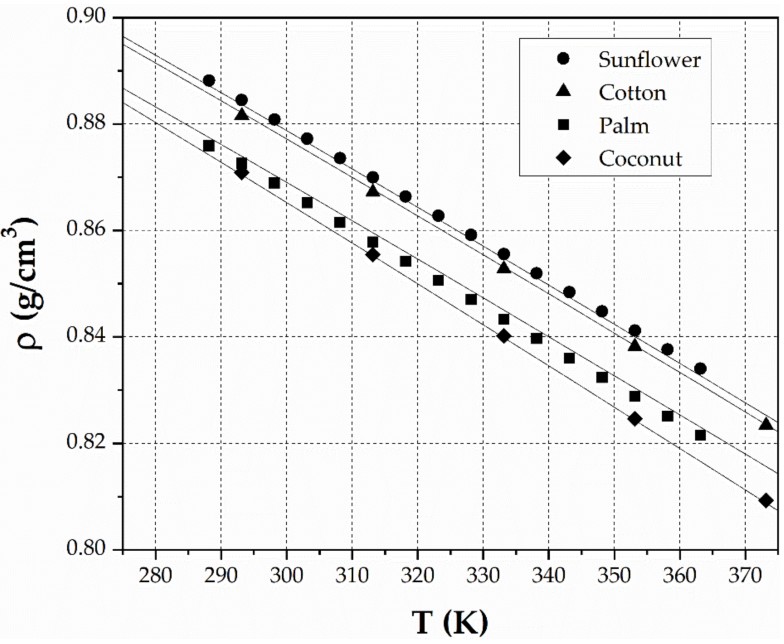

**Figure 5.** Density vs. temperature for some biodiesel samples.

Appendix B contains two examples to illustrate the use of the prediction methods presented in this work.

## 4. Discussion

First, we verified the data quality during its compilation, ensuring the experimental methods followed rigorous scientific procedures. We reviewed, for example, purity of substance, equipment calibration, reproducibility, and repeatability.

Importantly, we classified the prediction method verified here as semi-theoretical, since the experimental data were used to fit the GCVOL parameters.

The applicability and accuracy of predictions depend on their data origin and assumptions applied. Elbro et al., Ihmels and Gmehling, Pratas et al., and this work set up different assumptions. Elbro et al. derived their GCVOL parameters using solvents,

oligomers, and polymers [51], while Ihmels and Gmehling [52] tried to improve the predictions by proposing new GCVOL parameters, making an extension and revision of the parameters previously reported by Elbro et al. Ihmles and Gmehling incorporated the densities of various new compounds, among which are alkenes, cyclo-alkenes, alkynes, alcohols, carboxylic acids, fluorides, bromides, iodides, thiols, sulfides, sulfates, amines, nitriles, and nitro-compounds. Later, Pratas et al. predict the density of alkyl esters using parameters proposed by Elbro et al. for the groups -CH₃, -CH₂, and -COO-, except for the parameters for the =CH- group. Pratas et al. recalculated the parameters for the group =CH- using their experimental density information for different biodiesel samples. Those authors claimed that this change in the parameters for the =CH- effectively improves the prediction of unsaturated alkyl esters [8].

For our part, we estimated the GCVOL parameters using the densities of a high variety of alkyl esters, thus achieving an ad hoc characterization and better predictions. The GCVOL approaches were better to predict the alkyl ester density in the following order (from major to minor): This work > Ihmels > Pratas > Elbro. The above was based on the results for *AAD* and σ (Table 1). However, according to *R*, the order is This work > Pratas > Ihmels > Elbro (Table 1).

In short, deviations from −1 up to −5% and −1 up to −3.5% at temperatures higher than 365 °C (Figure 1) were found using the group contribution parameters proposed by Elbro et al. [51] and Pratas et al. [8]. In fact, that means slight overestimations of the experimental density in that range of temperature. From the above result, we postulate the "*C*" parameter (associated with the contribution of the $T^2$ term of Equation (3)) is required to reduce the error. For example, with the parameters proposed by Ihmels and Gmehling [52] or this work, we achieved lower and more dispersed errors.

Regarding the GRG optimization method, this has the disadvantage that it highly depends on the initial conditions. Indeed, the solver will most likely stop at the local optimum value, giving a solution that may or may not be optimized globally. Naturally, to deal with this issue, we probe different initial values to avoid reaching the same local solution. Furthermore, we selected the solution where our parameters were closer to the order of values reported in the literature.

We advertise that the prediction method tested here is applicable only to estimate the liquid density of alkyl esters. So, care must be taken at elevated temperatures or low molecular weights, conditions under which compounds may be present in the gas phase. Therefore, we recommend verifying the boiling point of compounds before the use of the present group contribution method.

Besides the comparison between the GCVOL approaches, we compare our results with the prediction obtained with the empirical correlation ($\rho = 1.069 + 3.575/MW + 0.0118N − 7.41 \times 10^{-4}T$) reported in our previous work [56] and with the approach based on an equation of state (EoS) [57]. We can observe that the GCVOL approach of the present work offers better predictions than the empirical correlation and the EoS model because *R* was closer to the unit, also *AAD*, and σ were lower when the GCVOL approach was used (Table 2). It is worth mentioning that the model based on the EoS has the drawback that when there is no information available in the literature on the critical properties of the compounds, it is necessary to estimate them, which accumulates calculation errors.

Regarding the effect that the type of alkyl ester has on the density, it can be observed that density depends on the number of times that the groups -CH₂ and =CH- are present in the alkyl ester. As the size of the alkyl ester increases, then the density decreases. The increase in the molecule size is directly related to the increase in their molecular weight, or it is also directly related to the increase in the number of times that the -CH₂ group is repeated in the alkyl ester. For a homologous series of saturated alkyl esters with the different number of carbon atoms, the groups -CH₃, =CH- and -COO- always remain constant with values of 2, 0, and 1 in the number of times that are present, while the number of times that the -CH₂ is present increases with the increase of the molecular size. For example, in the case of the methyl decanoate, methyl undecanoate, and methyl dodecanoate, their densities at the

temperature of 313.15 K are 0.8559, 0.8546, and 0.8534 g/cm$^3$, respectively [23]. That means that for each increase in one unit on the -CH$_2$ group, the density decreases by 0.00125 g/cm$^3$ on average. We described similar behavior in our previous work [56]. As the number of double bonds increases in the molecule, then the density increases. For each additional unsaturation, the =CH- group is repeated two times. For example, the density at 313.15 K of unsaturated alkyl esters with 19 carbon atoms and 1, 2, or 3 double bonds is 0.8595, 0.8715, and 0.887 g/cm$^3$, respectively [8], so it is observed that for each addition in the molecule, which translates into 2, 4 or 6 times that the group =CH- is repeated, the density increases between 0.012 to 0.0155 g/cm$^3$. The above coincides with what was described in our previous work. [56].

With biodiesel, various researchers have considered the role of the binary interaction parameters in Equation (5) as negligible, since it has been proven that very good approximations of the density of biodiesel are obtained, considering that the mixture is very close to an ideal solution because it comprises compounds of a similar chemical nature. However, to be more rigorous from a theoretical point of view, we should include the binary interaction parameters since always deviations from ideality exist, even if they are small.

It is worth mentioning that in terms of theoretical and thermodynamic fundamental considerations, we recommend the use of Equation (5) for prediction before Equation (6) because this equation considers deviations from the ideality of the mixtures. Unfortunately, to the best of our knowledge, there are no reported binary interaction values for the alkyl ester mixtures. Therefore, Equation (6) can be used considering the lack of binary interaction parameters.

An unexpected result was obtained when Equation (5) (considering $G_{ij} = 0$) and the parameters proposed by Pratas et al. [8] were tested because the biodiesel density was slightly better estimated instead of our approach. The above result is probably because Pratas et al. [8] adjusted the parameters of the =CH- group using the biodiesel densities instead of alkyl ester densities. We achieved the second-best result using Equation (6) and using the group contribution parameters derived in the present work (Table 3).

Equation (5) (considering $G_{ij} = 0$) has been widely used in the consideration of biodiesel as an ideal solution in the literature, giving good approximations. However, there always exist slight deviations from the ideality, because the interactions differ depending on the size and molecular geometry, or the presence or not of double bonds in the aliphatic chain of the alkyl esters. Deviations because of the non-ideality could be estimated through all binary interactions occurring between pairs of esters. The calculations of this work showed the importance of including binary interactions to improve prediction accuracy. It is worth mentioning that we encompass all the binary contributions through a correction factor. Therefore, future work is needed to estimate the binary interaction parameters for biodiesel.

The biodiesel composition depends on the raw material used for its production. Different vegetable oils or animal fats generate biodiesel with variations on alkyl ester composition, and, thus, with diverse density. Because the density of the biodiesel samples varies in a narrow area (Figure 4), it is difficult to establish behavior patterns. However, after analyzing the values of our database, we observed that the density increases with the increase of the alkyl esters that have a greater number of repeated -CH$_2$ and =CH- groups. For example, we can observe in Figure 5 that at any temperature, the density of the biodiesel samples increases in the following order from lower to higher values: coconut < palm < cotton < sunflower. This is because the average molecular weight of the coconut, palm, cotton, and sunflower biodiesel is 242.6088, 285.5073, 288.783, and 293.1052 g/mol, respectively, while the total concentration of unsaturated compounds, expressed in mass fraction, is 0.1903, 0.5503, 0.7273, and 0.8859, respectively.

It is worth mentioning that our prediction model applies to most biodiesel samples if they meet the following conditions: (1) the oils used to synthesize the biodiesel samples must be refined, ensuring that their composition is mostly by triacylglycerides, (2) the biodiesel samples must not contain hydroxyl groups in the aliphatic chain.

Therefore, our model must be used with caution with biodiesel samples made with waste cooking oil (WCO), because they could contain impurities causing deviations in predictions. In addition, we do not recommend using our model to predict the density of castor biodiesel, since it contains 80–90% of hydroxy fatty acid (methyl ricinoleate). Anyone can use our model with reliability to predict the density of both pure biodiesel and mixtures between them at any concentration. However, it does not apply to blends formed by biodiesel with fossil diesel.

## 5. Conclusions

First, a robust database for the densities of alkyl esters and biodiesel samples was assembled in this work, where a significant number of experimental data were compiled, organized, and classified. Next, new parameters for the GCVOL method were proposed ad hoc with the chemical compounds considered in the present study. Then, a complete comparison between results from the experiments and the model was executed in a wide range of temperatures. The most important result of this work was the corroboration of the high predictive accuracy of the model, which is applicable to biodiesel samples from diverse biomass origins. With the methodology presented here, we can predict the density of alkyl esters and biodiesel samples as a function of temperature with an AAD of 0.36 and 0.29%, respectively.

The method tested here could reduce the experimental determinations that are costly and time-consuming. In addition, we could easily incorporate this method into different simulation software. Finally, the capture and organization of the information was the most arduous task in this work. However, the high availability of data on the web facilitated our task.

**Supplementary Materials:** The following supporting information can be downloaded at: https://www.mdpi.com/article/10.3390/su14116804/s1, Table S1: Database for the alkyl ester density, Table S2: Fatty acid methyl ester composition of various biodiesel samples, Table S3: Database for the biodiesel density.

**Funding:** Mexican Petroleum Institute.

**Institutional Review Board Statement:** Not applicable.

**Informed Consent Statement:** Not applicable.

**Acknowledgments:** The author is grateful to the Mexican Petroleum Institute for its financial support through its Biomass Conversion Division. The author is grateful to Mirna Jimena Hernández Sánchez for her valuable help in compiling the information.

**Conflicts of Interest:** The author declares no conflict of interest.

## Appendix A

Table A1 shows the assignment that we made for the number of contribution groups present in some alkyl esters.

**Table A1.** Chemical structure and contribution groups of some alkyl esters.

| Compound | Chemical Structure | -CH$_3$ | -CH$_2$ | =CH- | -COO- |
|---|---|---|---|---|---|
| Methyl octadecanoate |  | 2 | 16 | 0 | 1 |
| Methyl cis-9-octadecanoate |  | 2 | 14 | 2 | 1 |
| Methyl (9Z,12Z)-octadeca-9,12-dienoate |  | 2 | 12 | 4 | 1 |
| Methyl (9Z,12Z,15Z)-octadeca-9,12,15-trienoate |  | 2 | 10 | 6 | 1 |

**Appendix B**

We show thought two examples to learn how to apply the prediction methods presented in this work. The first example illustrates the GCVOL method, and the second example exemplifies Kay's mixing rule.

Example 1. Calculate the methyl (9Z,12Z)-octadeca-9,12-dienoate (common name = methyl linoleate) density at 303.15 K using the group contribution parameters of this work and compare the result with the experimental value of 0.8792 g/cm$^3$ reported by Knothe and Steidley [23].

Solution.

The molecular weight of the methyl linoleate is calculated:

$MW_{\text{methyl (9Z,12Z)-octadeca-9,12-dienoate}}$ = [19(12.0107) + 34(1.00794) + 2(15.9994)] g/mol = 294.4721 g/mol.

The number of contribution groups for the methyl (9Z,12Z)-octadeca-9,12-dienoate are:

$n_{\text{-CH3}}$ = 2, $n_{\text{-CH2}}$ = 12, $n_{\text{=CH-}}$ = 4, $n_{\text{-COO-}}$ = 1 (See Table A1).

The methyl (9Z,12Z)-octadeca-9,12-dienoate molar volume is estimated by Equations (2) and (3), and using the group contribution parameters of Table 1:

$V_{\text{methyl (9Z,12Z)-octadeca-9,12-dienoate}}$ = {2[15.74 + 1.62 × 10$^{-3}$(303.15) + 10.01 × 10$^{-5}$(303.15)$^2$] + 12[14.42 + 5.1 × 10$^{-3}$(303.15) + 0.76 × 10$^{-5}$(303.15)$^2$] + 4[11.98 + 1.19 × 10$^{-3}$(303.15) + 0.89 × 10$^{-5}$(303.15)$^2$] + 1[30.77 + 1.31 × 10$^{-3}$(303.15) + 1.08 × 10$^{-5}$(303.15)$^2$]} cm$^3$/mol = 335.6289 cm$^3$/mol.

The methyl (9Z,12Z)-octadeca-9,12-dienoate density is calculated using Equation (1):

$\rho_{\text{methyl (9Z,12Z)-octadeca-9,12-dienoate}}$ = (294.4721 ÷ 335.6289) g/cm$^3$ = 0.87737 g/cm$^3$.

Then, the relative error is:

Error = [(0.8792 − 0.87737)100/0.8792]% = 0.09%.

Example 2. Given the composition of methyl esters for the palm biodiesel reported by Baroutian et al. [37]: $w_{\text{methyl hexadecanoate}}$ = 0.415, $w_{\text{methyl octadecanoate}}$ = 0.049, $w_{\text{methyl cis-9-octadecenoate}}$ = 0.401, $w_{\text{methyl (9Z,12Z)-octadeca-9,12-dienoate}}$ = 0.135, calculate its density at 303.15 K and compare the result with the experimental value of 0.86531 g/cm$^3$.

Solution.

The molecular weight of the alkyl esters is calculated:

(a) $MW_{\text{methyl hexadecanoate}}$ = [17(12.0107) + 34(1.00794) + 2(15.9994)] g/mol = 270.4507 g/mol.
(b) $MW_{\text{methyl octadecanoate}}$ = [19(12.0107) + 38(1.00794) + 2(15.9994)] g/mol = 298.5038 g/mol.
(c) $MW_{\text{methyl cis-9-octadecenoate}}$ = [19(12.0107) + 36(1.00794) + 2(15.9994)] g/mol = 296.4879 g/mol.
(d) $MW_{\text{methyl (9Z,12Z)-octadeca-9,12-dienoate}}$ = [19(12.0107) + 34(1.00794) + 2(15.9994)] g/mol = 294.4721 g/mol.

The number of contribution groups for the methyl esters are:

(a) $n_{\text{-CH3}}$ = 2, $n_{\text{-CH2}}$ = 14, $n_{\text{=CH-}}$ = 0, $n_{\text{-COO-}}$ = 1 (methyl hexadecanoate).
(b) $n_{\text{-CH3}}$ = 2, $n_{\text{-CH2}}$ = 16, $n_{\text{=CH-}}$ = 0, $n_{\text{-COO-}}$ = 1 (methyl octadecanoate).
(c) $n_{\text{CH3}}$ = 2, $n_{\text{-CH2}}$ = 14, $n_{\text{=CH-}}$ = 2, $n_{\text{-COO-}}$ = 1 (methyl cis-9-octadecenoate).
(d) $n_{\text{CH3}}$ = 2, $n_{\text{-CH2}}$ = 12, $n_{\text{=CH-}}$ = 4, $n_{\text{-COO-}}$ = 1 (methyl (9Z,12Z)-octadeca-9,12-dienoate).

The methyl ester molar volumes are estimated by Eqs. 2 and 3, and using the group contribution parameters of Table 1:

(a) $V_{\text{methyl hexadecanoate}}$ = {2[15.74 + 1.62 × 10$^{-3}$(303.15) + 10.01 × 10$^{-5}$(303.15)$^2$] + 14[14.42 + 5.1 × 10$^{-3}$(303.15) + 0.76 × 10$^{-5}$(303.15)$^2$] + 0[11.98 + 1.19 × 10$^{-3}$(303.15) + 0.89 × 10$^{-5}$(303.15)$^2$] + 1[30.77 + 1.31 × 10$^{-3}$(303.15) + 1.08 × 10$^{-5}$(303.15)$^2$]} cm$^3$/mol = 316.3233 cm$^3$/mol.

(b) $V_{\text{methyl octadecenoate}}$ = {2[15.74 + 1.62 × 10$^{-3}$(303.15) + 10.01 × 10$^{-5}$(303.15)$^2$] + 16[14.42 + 5.1 × 10$^{-3}$(303.15) + 0.76 × 10$^{-5}$(303.15)$^2$] + 0[11.98 + 1.19 × 10$^{-3}$(303.15) + 0.89 × 10$^{-5}$(303.15)$^2$] + 1[30.77 + 1.31 × 10$^{-3}$(303.15) + 1.08 × 10$^{-5}$(303.15)$^2$]} cm$^3$/mol = 349.6523 cm$^3$/mol.

(c)　$V_{\text{methyl cis-9-octadecenoate}}$ = {2[15.74 + 1.62 × $10^{-3}$(303.15) + 10.01 × $10^{-5}$(303.15)$^2$] + 14[14.42 + 5.1 × $10^{-3}$(303.15) + 0.76 × $10^{-5}$(303.15)$^2$] + 2[11.98 + 1.19 × $10^{-3}$(303.15) + 0.89 × $10^{-5}$(303.15)$^2$] + 1[30.77 + 1.31 × $10^{-3}$(303.15) + 1.08 × $10^{-5}$(303.15)$^2$]} cm$^3$/mol = 342.6406 cm$^3$/mol.

(d)　$V_{\text{methyl (9Z,12Z)-octadeca-9,12-dienoate}}$ = {2[15.74 + 1.62 × $10^{-3}$(303.15) + 10.01 × $10^{-5}$(303.15)$^2$] + 12[14.42 + 5.1 × $10^{-3}$(303.15) + 0.76 × $10^{-5}$(303.15)$^2$] + 4[11.98 + 1.19 × $10^{-3}$(303.15) + 0.89 × $10^{-5}$(303.15)$^2$] + 1[30.77 + 1.31 × $10^{-3}$(303.15) + 1.08 × $10^{-5}$(303.15)$^2$]} cm$^3$/mol = 335.6289 cm$^3$/mol.

The methyl ester densities are calculated using Equation (1):

(a)　$\rho_{\text{methyl hexadecanoate}}$ = [270.4507 ÷ 316.3233] g/cm$^3$ = 0.85498 g/cm$^3$.

(b)　$\rho_{\text{methyl octadecenoate}}$ = [298.5038 ÷ 349.6523] g/cm$^3$ = 0.85372 g/cm$^3$.

(c)　$\rho_{\text{methyl cis-9-octadecenoate}}$ = [296.4879 ÷ 342.6406] g/cm$^3$ = 0.8653 g/cm$^3$.

(d)　$\rho_{\text{methyl (9Z,12Z)-octadeca-9,12-dienoate}}$ = [294.4721 ÷ 335.6289] g/cm$^3$ = 0.87737 g/cm$^3$.

The palm biodiesel density is estimated using Equation (6) with $FC$ = 0.0054 g/cm$^3$:

$\rho_{\text{palm biodiesel}}$ = [0.415(0.85498) + 0.049(0.85372) + 0.401(0.8653) + 0.135(0.87737) + 0.0056] g/cm$^3$ = 0.86768 g/cm$^3$.

Then, the relative error is:

Error = [(0.86531 − 0.86768)100/0.86531]% = −0.27%.

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
