# Peer review of "A Group Contribution Method for Predicting the Alkyl Ester and Biodiesel Densities at Various Temperatures"

_sustainability, doi:10.3390/su14116804_

Round 1
Reviewer 1 Report
The work reported in this manuscript provides a group contribution method based approach to estimate the densites of alkyl esters, and Kay's mixing rule was applied to predict the densities of biodiesels and biodiesel blends. The group contribution parameters were arrived at based on measured data compiled from the published literature. The proposed approach was claimed to work well with an average absolute deviation of 0.29%. Further, the performace of the proposed approach was found to be better than the other Group Contribution based methods available in the literature. The manuscript provides useful data and approach for predicting biodiesel density at varying temperatures that can be used for biodiesel combustion modeling studies. The following comments require clarification from the author:
- It would be helpful, if some analysis on the effect of alkyl ester type and biodiesel type on the density is carried out using the proposed approach.
- How does the proposed group contribution based approach compare with the other available approaches in the literature to predict biodiesel density?
- Line 48: "Also, density plays a relevant role in fuel injection and diesel ignition". What role density has on ignition?
- Equation 5: Is the role of binary interation parameter significant in predicting biodiesel density?
- Which approach is recommended to predict the biodiesel density, is it binary interaction based (Eqn. 5) or correction factor based (Eqn. 6)? also provide justification for your recommendation.
- Table 1: It is important to provide information regarding the data base used to arrive at group contribution based parameters for the literature models. It plays a significant role in influencing the model predictions.
- Please comment on the similar values of group contribution parameters in Elbro et al. and Pratas et al. models.
- A careful proofread is required to correct the language errors. For instance, lines 52-53: "About the group contribution methods, it has developed efforts to predict the properties of alkyl esters".
- Editorial corrections: Lines 196-197: Figure 2a and Figure 2b, Figure 2: Improve the visibility of legends, Figure 4: Have different legends for different biodiesel samples.
Reviewer 2 Report
Authors did good research but according to my expertise more enhanced data is needed. As this journal focused on quality works. I suggest authors to improve the overall quality of manuscript and resubmit for consideration. Comments: 1. Authors divide is sentence in each paragraph. 2. Introduction is too short. 3. Methods part too short. 4. Authors used the graphs are drawn with demo version. That makes not advisable.Author Response
Please see the attachment.

Reviewer 3 Report
Good work on predicting the quality of biodiesel. Here are my comments:
1] Clarify whether this prediction model is applicable for all kind of biodiesel (eg. palm biodiesel, soybean biodiesel, karanja biodiesel, WCO biodiesel)
2] Is it applicable for blend biodiesel (eg. B5, B7, B10). Explain. Since biodiesel in blend form are common, justify the applicability of the findings of this work on blends.
3] The discussion can be improved by providing more explanations. Findings must be supported by the theoretical explanation, wherever possible.
Round 2
Reviewer 2 Report
Accept.